# Intimate Dynamics and Relationship Satisfaction among LGB Adolescents: The Role of Sexual Minority Oppression

**DOI:** 10.3390/children8030231

**Published:** 2021-03-17

**Authors:** Henrique Pereira, Graça Esgalhado

**Affiliations:** 1Department of Psychology and Education, Faculty of Social and Human Sciences, University of Beira Interior, Pólo IV, 6200-209 Covilhã, Portugal; mgpe@ubi.pt; 2Health Sciences Research Centre (CICS-UBI), 6200-506 Covilhã, Portugal; 3Research Centre in Sports Sciences, Health Sciences and Human Development (CIDESD), 5001-801 Vila Real, Portugal; 4Institute of Cognitive Psychology, Human and Social Development (IPCDHS), 3000-115 Coimbra, Portugal

**Keywords:** adolescence, intimate dynamics, relationship satisfaction, young LGB couples, sexual minority oppression

## Abstract

Adolescent lesbian, gay, and bisexual (LGB) romantic partners face the challenge of developing satisfactory relationships while managing stressors associated with being members of a stigmatized minority group due to their sexual minority status. The aims of this study were to explore and describe relationship dynamics among LGB adolescents that are in committed same-sex relationships in Portugal, to assess levels of satisfaction with their relationships, and to assess whether LGB oppression was associated with the likelihood of anticipating and experiencing problems within the relationship. A sample of 182 self-identified LGB adolescents (mean age = 17.89 years; SD = 1.99), completed an online survey consisting of various sociodemographic measures, a relationship dynamics questionnaire, a self-assessment of relationship satisfaction, and an adapted version of the Gay and Lesbian Oppressive Situations Inventory. Results show that participants were highly satisfied with their relationships, except those who were non-monogamous and less committed to their relationships. Participants showed moderate levels of LGB oppression situations, and, as demonstrated by the hierarchical multiple regression analysis, age (being older), having lower levels of commitment, and being exposed to higher levels of exclusion, rejection, and separation were strong predictors of lower levels of relationship satisfaction.

## 1. Introduction

Lesbian, Gay, and Bisexual (LGB) individuals are often stigmatized as being a minority, with same-sex relationships not being supported or recognized in the same way by society as are heterosexual relationships [1,2], influencing how individuals satisfy their needs and goals of intimacy. The experience of stigmatization can be a major factor in social stress, which leads to decreased relationship quality and satisfaction, increased conflicts, loneliness, and sexual problems [3,4]. The risks and challenges faced by LGB people who belong to marginalized communities are well documented in the literature [5,6,7,8,9], and LGB individuals are found to be more susceptible to physical and mental illness, suicide, and stress, with adolescent LGB people being at a greater risk.

Research has corroborated the existence of risks for LGB adolescents, both in cases where young people come out or in cases where they keep their sexual orientation hidden [10]. If, on the one hand, peer relationships play an important role in the lives of young people and adolescents, regardless of their sexual orientation, on the other hand LGB youth and adolescents have more serious general health outcomes and well-being issues when compared with their heterosexual peers [11,12,13]. In this sense, LGB young people are more likely to find mechanisms to protect against harassment and victimization [14,15], such as establishing significant relationships, since these strategies would allow them to moderate or minimize the impact of internalized homonegativity [15]. 

Regarding relationship dynamics, the fact that LGB youth have greater difficulty in gaining support from their parents [16] creates the expectation that significant relationships constitute an important source of social support, even if they may take place in privileged contexts, such as at school or on the Internet [17]. In fact, the Internet is a useful tool for young people to access information, in addition to assisting in finding information about offline services, such as support groups, clinics, or community centers. The Internet can also help young people find friends and romantic partners, with anonymity being an important factor in these online contacts, as it gives young people a sense of protection from the stigma surrounding homosexual practices [18,19].

It is clear that most human beings seek to explore their intimacy through satisfactory romantic relationships [20], and these are fundamental to human development [21]. For LGB youth in particular, adolescence presents some peculiar challenges, as it is a sensitive period of exploration of their sexuality and the onset of romantic relationships [17]. Engagement and satisfaction in romantic relationships is a significant event in the transition to adulthood and can present a number of mental and physical health challenges [22]. Amidst the exploration of their sexual identity, LGB adolescents may fear being judged, excluded, or oppressed because of their sexual orientation [23] which may result in a process of concealment of their sexual identity, internalization of homonegativity, and invalidation of romantic same-sex relationships [24].

Several studies show that reaching and maintaining satisfactory relationships may be a difficult task for LGB adolescents because, when compared to heterosexual adolescents, LGB youth are at a higher risk for mental health impairment, in addition to being more likely to experience difficulties at school [25]. LGB adolescents who are also part of a racial/ethnic minority are exposed to prejudice and discrimination based on race/ethnicity, as well as prejudice and discrimination within their ethnic/racial communities for being LGB and are more likely to be at even greater risk [26,27]. Teenagers with same-sex attraction may be more depressed or experience lower self-esteem due to internalized homonegativity and worry about finding a romantic partner; they also have doubts about having a family of their own, or always expect to feel different from their peers. These factors may lead to depression and low self-worth even if these young adults’ relationships with their family, peers, and mentors are not negatively affected by their feelings of same-sex attraction [28]. Feelings of loneliness, suicide attempts [9] and the fact that they are verbally and physically vulnerable to attack, are all factors that may contribute to an increase in substance abuse [29] and overall health impairment. 

Exposure to LGB oppression and social discrimination are highly important components associated with the establishment of significant relationships among LGB youth. Although there are very few studies that document the extent to which LGB youth are at risk of being victims of violence, it is estimated that between 40% to 80% of young people have suffered from violence associated with their sexual minority status [30]. Therefore, the identity development of LGB youth is restricted by these negative societal attitudes, which may perpetuate the experience of stigmatization and discrimination, findings that are consistent with the minority stress theory [31], suggesting that stigma and interpersonal discrimination are risk factors for physical and mental health problems among sexual minorities [32]. Nevertheless, little information exists on how exposure to social discrimination may impact the quality of relationships among LGB adolescents. 

Being in a primary relationship is a common experience shared by many adults, but, for adolescents and young adults, individuals bring with them a learning history that shapes their behavioral repertoire and expectations, conditioned by social beliefs consistent with the expectation of rejection. Since individuals belonging to a sexual minority face high rates of mental health issues, it is likely that this will affect the quality of their interpersonal relationships [33]. The existing literature is characterized by a lack of relationship dynamics knowledge among LGB youth. Furthermore, literature that examines the correlation between mental and physical health and social discrimination [8] does not specifically examine such associations in young couples belonging to a sexual minority. Less attention has been paid to other aspects of one’s relationship history (the onset of dating behavior, relationship length, etc.). These types of relationship milestones may be important in understanding how social discrimination becomes linked to primary partner relationship quality and satisfaction. 

In Portugal, although there is a non-discrimination clause based on sexual orientation in the Portuguese Constitution, a law that allows same-sex couples from 16 years of age to marry since 2011, and a law that allows adoption and joint-adoption of children by same-sex couples since 2016, there are still discriminatory practices in society mainly due to Catholic religious conservatism that is embedded in the country [8,15]. Therefore, Portuguese LGB adolescents are thus exposed to ambivalent circumstances of both political inclusion and social oppression that are likely to have a significant impact on their quality of life.

The following overarching research questions were posed: (1) what are the relationship dynamics among LGB adolescents?; (2) what are the levels of relationship satisfaction (general and across comparison groups)?; (3) what are the levels of experienced LGB oppression?; (4) what are the levels of association between LGB oppression and relationship satisfaction?; and (5) what are the predictors of relationship satisfaction among LGB adolescents?

## 2. Materials and Methods

Sociodemographic Questionnaire. Items included age, gender, sexual orientation, marital status, occupational status, education, and economic status.

Intimate Dynamics. Items included: the duration of relationship (in months), cohabitation, location of where the couple first met, types of discrimination experienced (if any), future plans to have children, relationship arrangements (monogamy or other arrangements), relationship problems (if any), support seeking, level of commitment, major sources of support as a couple, and the number of weekly sexual interactions.

Relationship Satisfaction. This was a single item measure. The respondents were asked to estimate their current levels of satisfaction with their relationship on a scale from one to ten (1 = totally unsatisfied; 10 = completely satisfied).

Gay and Lesbian Oppressive Situations Inventory. The Gay and Lesbian Oppressive Situations Inventory—Frequency (GALOSI–F) [34] is a 49-item survey, which measures the frequency of perceived heterosexist and antigay discrimination faced by gay and lesbian individuals across a variety of settings. Each item uses a 5-point Likert-type scale ranging from 1 (never) to 5 (almost always). The measure is composed of seven GALOSI–F scales; however, only 15 items from four subscales were used in this study. The first item is the Couples’ Issues Scale (four items—I have been uncomfortable about introducing my partner/girlfriend/boyfriend to family members; I have seen that it is harder for gays to have children than it is for heterosexuals; I have been uncomfortable about bringing my partner/girlfriend/boyfriend to work-related social events; I have been afraid to publicly display affection with my partner/girlfriend/boyfriend). Secondly, the study utilized the Dangers to Safety Scale (two items—I have been physically threatened because of my sexuality; I have thought about committing suicide). Thirdly, the study incorporated the Exclusion, Rejection, and Separation Scale (five items—People have avoided me because of my sexual orientation; I have felt isolated by members of my family because of my sexual orientation; I have been afraid that my family would reject me because of my sexual orientation; My family has denied the existence of my sexual orientation; I have had family members ask me to pretend that I am not gay). Finally, the study used the Internalized Homonegativity Scale (4 items—My sexual orientation has been in conflict with my religious beliefs; It has been hard for me to feel good about myself because of people’s negative views about my sexual orientation; It has been hard for me to accept my sexual orientation; I have felt depressed concerning my sexual orientation). Subsequently, we tested this survey and found very good internal consistency reliability (Cronbach alpha = 0.88).

Data collection was aimed at self-identified LGB adolescents living in Portugal. The internet was used as a means of disseminating measurement instruments, through notifications sent to LGB youth organization, social networks, such as Facebook, and mailing lists. Participation was voluntary. Participants were referred to a linked website created specifically for the purposes of this investigation. The first page of the questionnaire explained the objectives of the study, and informed participants about how to fill it in, how to withdraw from the study, and how to contact the authors for more information. They were also asked to read and agree to an informed consent waiver.

A total of about 1000 notifications were sent and 182 participants responded voluntarily (20% response rate). The dissemination of the survey complied to all of the ethical principles of informed consent, anonymity and confidentiality. Neither rewards nor other incentives were offered. Inclusion criteria included the following: being younger than 20 years of age, being a Portuguese native speaker, and being in a same-sex committed relationship. Ethical approval for this study was granted by the Ethics Committee of the University of Beira Interior, Portugal (code CEUBI-Pj-2020-088).

Data analysis. Descriptive statistics were performed to describe the sample (mean, standard deviation, frequencies and percentages). To evaluate levels of satisfaction with relationships and levels of LGB oppression, descriptive statistics were also used. To evaluate whether there were differences between comparison groups, Student *t*-tests, and one-way ANOVAs were conducted. To assess the association between relationship satisfaction and LGB Oppression, Pearson Correlation Coefficients were conducted. Finally, a hierarchical linear regression analysis was conducted to examine the effects of independent variables (Age, Gender, Sexual orientation, Length of relationship, Relationship arrangement, Level of commitment, Couple’s issues, Dangers to safety, Exclusion, rejection, and separation, and Internalized homonegativity) on the dependent variable (Relationship Satisfaction). To avoid type I errors, Bonferroni correction tests were run.

## 3. Results

### 3.1. Sociodemographic Characteristics

A total of 182 self-identified adolescent LGB participants between 16 and 20 years, with a mean age of 17.89 (*SD* = 1.99) years, completed an online survey. Demographic data (Table 1) show that the sample members are highly educated and that the sample is composed of participants who self-identify as gay, bisexual, or lesbian. The majority of the participants possess a complete secondary or university education, all are in a stable relationship, despite the fact that only 17% are married or in a civil union/cohabitation, and 95% live in urban centers.

### 3.2. Intimate Dynamics

Descriptive statistics regarding the intimate dynamics of LGB adolescents are presented in Table 2. On average, they have been together as a couple for almost 15 months, but the majority do not live together. Most state that they met via the internet (30%), through friends (18%) or at school or university (26%). Interestingly, 42% said that they did not feel any type of discrimination as a couple; whereas, 30% felt it from their own family members. The majority of our participants say that they are monogamous (76%), plan on having children (58%), and do not seek support when in need (84%). Regarding relationship problems, the main topics mentioned were jealousy (68%), lack of or poor communication (66%), and difficulties in managing friendships (54%). Only 2% reported problems related to physical or sexual violence. Overall, 36% of participants stated that they were committed to their present relationship for life, and 44% for the long-term. Major sources of social support as a couple included straight friends (76%), gay friends (60%), lesbian friends (52%), other LGB couples (42%), and their mothers (40%). Finally, only 22% reported having sexual problems and, on average, they had sex 3.17 times per week (*SD* = 2.28).

### 3.3. Levels of Satisfaction

Results show that young LGB participants were highly satisfied with their relationships (*M* = 8.10; *SD* = 1.58). Despite the fact that male participants were more satisfied than female participants, and that gay participants were more satisfied than lesbian and bisexual participants, these differences were not statistically significant. Yet, when comparing levels of satisfaction between monogamous and non-monogamous participants, results showed that monogamous couples were more satisfied (*M* = 8.41; *SD* = 1.27) than non-monogamous couples (*M* = 6.33; *SD* = 1.15). These differences were statistically significant [*t*(40) = 2.730; *p* = 0.008].

We also found significant results regarding the levels of satisfaction within their relationships when comparing groups by level of commitment [*F*(2;47) = 16.287; *p* < 0.001]. Participants who expected to remain in the same relationship for life were more satisfied (*M* = 8.67; *SD* = 0.84) than those who expected to be in their relationship for the long-term (*M* = 8.55; *SD* = 0.96), and substantially more satisfied than participants who only expected to be in their relationship for the time being (*M* = 6.10; *SD* = 2.13).

Finally, we also looked into the differences in levels of satisfaction between participants who reported being in a relationship for shorter or longer periods of time, using the median (8.5 months) as the cut-off point. In regard to relationship duration, we also found significant results, indicating that participants who have been in a relationship for a longer period of time were more satisfied (*M* = 8.54; *SD* = 1.06) than participants who reported shorter relationship periods (*M* = 7.75; *SD* = 1.92) [*t*(46) = −1.770; *p* = 0.043]. All results are shown in Table 3.

### 3.4. Levels of Oppression

Oppression was measured for each of the GALOSI-F scales used in this study, as well as for total levels of oppression found in the sum of all 15 items of the questionnaire. Scores were compared to the expected median for each scale. Therefore, overall levels of oppression were relatively low (*M* = 39.72; *SD* = 13.65) compared to the expected median (*M_dn_* = 45). This was also the case for all other scales, which included couple’s issues (*M* = 11.66; *SD* = 3.56/*M_dn_* = 12), dangers to safety, (*M* = 3.82; *SD* = 2.18/*M_dn_* = 6), exclusion, rejection, and separation (*M* = 13.46; *SD* = 5.91/*M_dn_* = 15), and internalized homonegativity (*M* = 7.36; *SD* = 3.40/*M_dn_* = 12).

Table 4 and Table 5 show the comparison of levels of LGB oppression by gender, sexual orientation, the type of relationship agreement, the expected length of the relationship, and the actual length of the relationship. Results showed statistically significant differences for general levels of oppression by the type of relationship agreement [*t*(40) = −2.813; *p* = 0.008], indicating that non-monogamous participants experienced more oppression (*M* = 59.00; *SD* = 12.76) than monogamous participants (*M* = 37.82; *SD* = 12.55). When examining the differences for each dimension, results showed that for “couple’s issues”, non-monogamous participants reported higher levels of oppression (*M* = 16.33; *SD* = 1.52) than monogamous participants (*M* = 11.17; *SD* = 3.50) [*t*(40 = −2.508; *p* = 0.016]. In addition, in regard to “couple’s issues”, participants who expected to remain in their relationships only for the time being also showed higher levels of oppression (*M* = 13.90; *SD* = 2.64) than those who expected to be in their relationship for a lifetime (*M* = 10.83; *SD* = 3.01) or for the long-term (*M* = 11.31; *SD* = 4.01) [*F*(2;47) = 2.748; *p* = 0.034]. No significant differences were observed for the “dangers to safety” or “exclusion, rejection, and separation” dimensions. Results indicated significant differences for the type of relationship agreement and for the expected length of the relationship. Non-monogamous participants scored higher on this scale (M = 19.33; SD = 5.80) than monogamous participants (*M* = 13.23; *SD* = 5.80) [*t*(40 = −1.746; *p* = 0.049], and couples who expected to remain in their relationship for the time being scored higher (*M* = 16.70; *SD* = 5.12) than those who expected to be in their relationship for a lifetime (*M* = 11.66; *SD* = 5.46) or for the long-term (*M* = 13.45; *SD* = 6.20) [*F*(2;47) = 2.464; *p* = 0.046].

### 3.5. Relationship Satisfaction and Oppression

As shown in Table 6, significant negative correlations between Relationship Satisfaction and LGB Oppression were obtained for the scales measuring “internalized homonegativity” (*r* = −0.309; *p* = 0.029) and “overall levels of oppression” (*r* = −0.286; *p* = 0.044). The same table also demonstrates that the total levels of oppression were very strongly and positively correlated with other dimensions of oppression, especially “internalized homonegativity” (*r* = 0.843; *p* ≤ 0.001), “exclusion, rejection and separation” (*r* = 0.835; *p* < 0.001), and “couple’s issues” (*r* = 0.797; *p* < 0.001).

A hierarchical multiple regression analysis was performed to assess the effects of oppression on relationship satisfaction. Possible confounding variables that were added in the first block included age, gender, sexual orientation, the length of the relationship, the type of relationship arrangement, and the level of commitment. Dimensions of the GALOSI–F scale were added in the second block, which encompassed couple’s issues, dangers to safety, exclusion, rejection, and separation, and internalized homonegativity. The first block of the analysis explained 32% of the overall variance, while the second block—dimensions of oppression—explained an additional 5%. As shown in Table 7, age and levels of commitment were significant predictors of relationship satisfaction. In the second step, age, levels of commitment, and the exposure to exclusion, rejection, and separation were found to be significant predictors.

## 4. Discussion

Adolescent same-sex romantic partners face the challenge of developing positive relationships while managing stress associated with being members of a stigmatized minority group. This study examined the intimate dynamics of adolescent LGB couples’ relationship satisfaction, and more specifically the link between sources of oppression that might influence relationship satisfaction, as addressed by our research questions. The results of this study show that adolescent LGB couples are capable of maintaining well-functioning and stable romantic relationships, despite the fact that they are exposed to various forms and levels of discrimination, with these results being consistent with the previous literature findings [35].

Much of the research on social stigma and romantic relationship functioning among sexual minorities has neglected to examine other relevant social identities, such as age. Our results examine several important dynamics of adolescent LGB couples, such as future plans to have children, being monogamous, or possessing relationship problems that are common to older LGB and heterosexual couples [36]. Yet some of these dynamics may encompass the insidious influence of social isolation due to the expectation of rejection, including examples such as being spurned by one’s own family, or not seeking support when in need. Perhaps this is one of the most significant differences between adolescent same-sex couples and adolescent heterosexual couples, as same-sex romantic partners must often forge and maintain romantic bonds in social realms that frequently marginalize and devalue same-sex relationships [37]. In this study, however, the anticipation of discrimination and experiencing actual discrimination were associated with heightened levels of relationship satisfaction.

High levels of relationship satisfaction may result from one partner’s use of the other as a primary source of emotional support, intimacy, trust, and communication [38]. However, couples may also feel that their communication is overly restricted to the relationship, particularly if they are unable to articulate their emotional needs with others, especially when facing possible adversity due to social discrimination [39]. In fact, research has identified this type of emotional and social support as a buffer against the negative effects of discrimination on mental health [40,41], a finding that appears to be confirmed by our results. 

Our results also confirm that nearly all our participants are exposed to some level of oppression, but that non-monogamous and short-term couples are particularly vulnerable to its negative effects. These couples scored lower in all dimensions of oppression, and it may be consistent with the trend of younger LGB generations being seemingly more inclined towards monogamy than their elders [42]. On the other hand, this may indicate a coping mechanism to deal with oppression, since the decision to be non-monogamous may include other inherent factors, such as, self-exclusion, anticipation of rejection, or parental disapproval. These factors may be present in addition to the advantage of having an increased number of sexual partners [43].

Couples that expected their relationship not to last for very long may be facing relationship problems, which, in turn, could directly affect their relationship satisfaction. Our results demonstrate that these couples also score higher on levels of oppression, leading us to infer that oppression negatively interferes with relationship satisfaction. This trend was also found in our correlational results, which were particularly affected by internalized homonegativity. Specifically, young LGB participants with higher levels of internalized homonegativity are more likely to hide their sexual orientation and relationship status in different social contexts [15,44]. In this sense, the results suggest that young LGB couples experience heightened levels of the three main stressors that compose the minority stress model, defined as internalized homonegativity, stigma, and discrimination [31], all of which predict lower levels of relationship satisfaction. This finding should be understood in light of the fact that participants may undervalue their same-sex relationship in order to avoid being exposed to sexually-based discrimination and violence.

Furthermore, this research shows that oppression has negative impacts on adolescent LGB couples in terms of relationship satisfaction. As demonstrated by the hierarchical multiple regression analysis, age (being older), having lower levels of commitment, and being exposed to higher levels of exclusion, rejection, and separation were strong predictors of lower levels of relationship satisfaction. Older participants may be less satisfied with their relationships due to the fact that they may be more likely to be exposed to external processes, such as family stressors, and their lower expectations of remaining in the relationship may be influenced by obstacles in managing commitment and negotiation [22]. Despite the fact that existing stage models do not adequately capture adolescent LGB couples’ relationship experiences and levels of satisfaction, our results are consistent with a growing body of literature confirming the negative effect of exclusion on relationship quality among LGB young people [45,46], even if young LGB couples have to deal with the same challenges heterosexual couples face at the level of general relationship functioning [47].

It cannot be forgotten that this study was carried out in the cultural context of Portuguese society, a country where there are political and social measures that contribute to a safer environment, allowing adolescents to have access to positive models for the expression of their sexual identity, through the recognition of LGB identities by the State [48]. In fact, in Portugal there is legislation that regulates same-sex marriages and the adoption by same-sex couples, for example, which has been an important contribution to the success of the valorization of sexual minority identities, despite heterosexism, personal intolerance and the discrimination that still exists in Portuguese society [49]. Thus, this study contributes to a better understanding of the positive influences for LGB adolescents, highlighting the importance of significant relationships, family and friend’s social support [50], reinforcing the relevance of positive social environments where adolescents live, and which can directly affect their overall well-being [51].

This study is part of a small but growing body of research documenting the adverse effects of oppression on young same-sex couples’ relationship dynamics and satisfaction, suggesting that the well-being of adolescent LGB couples can be supported by efforts to decrease negative societal messages about young LGB persons and increase positive images of young same-sex couples. A social climate that is more affirming of young same-sex couples will be likely to reduce the self-stigmatization and vulnerability to oppression that has been linked to lower levels of relationship satisfaction in this study.

This study is not without limitations. The convenience sample used was recruited via the Internet, and therefore the results cannot be generalized. Future endeavors should be conducted using larger and more representative samples. Also, relationship satisfaction was measured using a single ordinal item. Although we believe that this measurement was adequate to evaluate the self-assessment of relationship satisfaction, future studies should include more complex and robust measures of relationship functioning. Finally, the fact that the sample was recruited online through contacts with various LGB associations and interest groups targeting LGB youth may have resulted in a degree of homogeneity in the results, associated with the availability of access to information about the issues facing sexual minorities, in addition to online and offline sources of social support.

## 5. Conclusions

In light of the present findings, it is perhaps remarkable that young same-sex couples appear to be so highly satisfied with their relationships, given the added burden of oppression with which they must cope. A lesson to be learned from these results is that young LGB couples may actually have high levels of resilience factors associated with establishing significant relationships that protect them and their relationships from the negative effects of sexual stigma. This research shows that oppression has negative impacts on young LGB people in terms of their relationship satisfaction. Therefore, positive measures to promote respect for the human rights of young lesbian, gay, and bisexual people in relationships should be adopted and implemented to minimize the adverse consequences of prejudice on young couples, in addition to providing insight about ways that couples can become stronger by successfully facing adversity.

## Figures and Tables

**Table 1 children-08-00231-t001:** Sociodemographic Characteristics (*N* = 182).

Variable	Category	*N*	%
Gender	Male	80	44%
	Female	102	56%
Sexual Orientation	Gay	76	42%
	Lesbian	62	34%
	Bisexual	44	24%
Marital Status	Single	151	83%
	Married	3	2%
	Civil Union	9	5%
	Cohabitation	19	10%
Occupational Status	Student	141	77.4%
	Employed	41	22.6%
Education	Complete Primary	11	6%
	Complete Secondary	95	52%
	College/University Degree/Attendance	76	42%
Economic Status	Low	62	34%
	Average	98	54%
	High	22	12%
Place of residence	Urban	167	92%
	Rural	15	8%

**Table 2 children-08-00231-t002:** Results for Relationship Dynamics.

Variables	Categories	*N*	%
How they Met	Through friends	33	18%
	At school/university	47	26%
	Through the internet	55	30%
	At a social event	15	8%
	Other	32	18%
Type of Discrimination Felt	None	76	42%
	Family rejection	55	30%
	Other	51	28%
Planning to Have Children	Yes	106	58%
	No	76	42%
Monogamous	Yes	138	76%
	No	44	24%
Relationship Problems (multiple options)	Jealousy	124	68%
	Communication	120	66%
	Friendships	98	54%
	Lack of trust	55	30%
	Sexual problems	40	22%
	Changes in love	40	22%
	Financial problems	33	18%
	Health problems	29	16%
	Verbal violence	25	14%
	Physical violence	4	2%
	Sexual violence	4	2%
Support Sought	Yes	29	16%
	No	153	84%
Level of Commitment	For life	66	36%
	For the long-term	80	44%
	For the time being	22	12%
	For a short period of time	14	8%
Major Sources of Support as a Couple (multiple options)	Straight friends	138	76%
	Gay friends	109	60%
	Lesbian friends	95	52%
	Other gay or lesbian couples	76	42%
	Mother	73	40%
	Brothers or sisters	58	32%
	Community groups	47	26%
	Other family members	33	18%

**Table 3 children-08-00231-t003:** Results for Levels of Satisfaction Across Comparison Groups.

Variables	Categories	*M*	*SD*	*t/F(df)*	*p*
Level of Satisfaction with the Relationship	-	8.10	1.58	-	-
Level of Satisfaction by Gender	Male	8.41	1.08	1.232 (48)	0.228
	Female	7.86	1.90		
Level of Satisfaction by Sexual Orientation	Gay	8.45	.99	0.438 (2;46)	0.648
	Lesbian	8.18	1.70		
	Bisexual	7.92	1.24		
Level of Satisfaction by Relationship Arrangement	Monogamous	8.41	1.27	2.730 (40)	0.008 *
	Non-monogamous	6.33	1.15		
Level of Satisfaction by Level of Commitment	For life	8.67	0.84	16.287 (2;47)	0.000 **
	For the long-term	8.55	0.96		
	For the time being	6.10	2.13		
Level of Satisfaction by Length of Relationship (Median = 8.5 months)	Shorter length	7.75	1.92	−1.770 (46)	0.043 *
	Longer length	8.54	1.06		

* < 0.05; ** < 0.001.

**Table 4 children-08-00231-t004:** Results for couple’s issues, dangers to safety and exclusion, rejection and separation by gender, sexual orientation and relationship characteristics.

Variables	Categories	Couple’s Issues		Dangers to Safety		Exclusion, Rejection, and Separation	
		*M(SD)*	*t/F(df)/p*	*M(SD)*	*t/F(df)/p*	*M(SD)*	*t/F(df)/p*
Gender	Male	11.77 (3.65)	0.196 (48)/0.845	4.31 (2.27)	1.447(48)/0.154	13.72 (5.64)	0.280(48)/0.780
	Female	11.57 (3.55)		3.42 (2.06)		13.25 (6.22)	
Sexual Orientation	Gay	11.55 (3.84)	0.022 (2;46)/0.978	4.30 (2.25)	1.009(2;46)/0.373	13.75 (5.73)	1.968(2;46)/0.151
	Lesbian	11.64 (3.95)		3.82 (2.37)		15.17 (6.23)	
	Bisexual	11.83 (2.85)		3.16 (1.74)		10.83 (5.40)	
Type of Relationship Arrangement	Monogamous	11.17 (3.50)	−2.508 (40)/.016 *	3.71 (2.12)	−0.222 (40)/0.825	13.23 (5.80)	−1.746(40)/0.049 *
	Non-monogamous	16.33 (1.52)		4.00 (2.00)		19.33 (6.42)	
Expected Length of Relationship	For life	10.83 (3.01)	2.748 (2;47)/0.034 *	3.83 (2.66)	0.019(2;47)/0.981	11.66 (5.46)	2.464(2;47)/0.046 *
	For the long-term	11.31 (4.01)		3.86 (2.03)		13.45 (6.20)	
	For the time being	13.90 (2.64)		3.70 (1.70)		16.70 (5.12)	
Length of Relationship	Shorter length	11.20 (3.87)	−0.754 (46)/0.454	3.87 (2.00)	−0.065(46)/0.948	12.08 (5.50)	−1.460 (46)/0.151
	Longer length	12.00 (3.37)		3.91 (2.41)		14.58 (6.33)	

* < 0.05.

**Table 5 children-08-00231-t005:** Results for internalized homonegativity and general LGB oppression by gender, sexual orientation and relationship characteristics.

Variables	Categories	Internalized Homonegativity		Overall LGB Oppression	
		*M(SD)*	*t/F(df)/p*	*M(SD)*	*t/F(df)/p*
Gender	Male	7.40 (3.47)	0.089(48)/0.929	40.63 (13.95)	0.417(48)/0.678
	Female	7.32(3.41)		39.00 (13.62)	
Sexual Orientation	Gay	7.35 (3.49)	0.034(2;46)/0.967	40.45 (14.29)	0.438(2;46)/0.648
	Lesbian	7.41 (3.80)		41.29 (14.73)	
	Bisexual	7.08 (2.93)		36.58 (12.02)	
Type of Relationship Arrangement	Monogamous	6.71 (3.05)	−2.685(40)/0.011 *	37.82 (12.55)	−2.813 (40)/0.008 *
	Non-monogamous	11.66 (3.51)		59.00 (12.76)	
Expected Length of Relationship	For life	6.88 (3.69)	2.124(2;47)/0.131	36.61 (14.15)	2.051 (2;47)/0.140
	For the long-term	6.86 (2.93)		38.90 (13.09)	
	For the time being	9.30 (3.46)		47.10 (12.44)	
Length of Relationship	Shorter length	7.16 (3.61)	−0.172 (46)/0.864	37.70 (15.23)	−0.889 (46)/0.379
	Longer length	7.33 (3.08)		41.25 (12.20)	

* < 0.05.

**Table 6 children-08-00231-t006:** Correlational Matrix for the Results Associating LGB Oppression and Relationship Satisfaction.

	1	2	3	4	5	6
1—Couple’s Issues	1					
2—Dangers to Safety	0.289 *	1				
3—Exclusion, Rejection, and Separation	0.507 **	0.449 **	1			
4—Internalized Homonegativity	0.694 **	0.416 **	0.514 **	1		
5—Overall Levels of Oppression	0.797 **	0.602 **	0.835 **	0.843 **	1	
6—Relationship Satisfaction	−0.258	−0.151	−0.203	−0.309 *	−0.286 *	1

* < 0.05; ** < 0.001.

**Table 7 children-08-00231-t007:** Hierarchical Multiple Regression Analysis Predicting Relationship Satisfaction.

Predictor		*R^2^*	*β*	*p*
Step 1		0.322		0.012 *
	Age		−0.308	0.048 *
	Gender		−0.064	0.725
	Sexual orientation		−0.113	0.521
	Length of relationship		0.152	0.302
	Relationship arrangement		−0.130	0.350
	Level of commitment		−0.530	0.001 *
Step 2		0.366		0.043 *
	Age		−0.420	0.031 *
	Gender		−0.040	0.831
	Sexual orientation		−0.195	0.304
	Length of relationship		0.243	0.146
	Relationship arrangement		−0.156	0.288
	Level of commitment		−0.476	0.005 *
	Couple’s issues		0.018	0.930
	Dangers to safety		−0.019	0.910
	Exclusion, rejection, and separation		−0.298	0.046 *
	Internalized homonegativity		0.159	0.486

* < 0.05.

## Data Availability

The data presented in this study are available upon request.

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
