# Peer review of "Intimate Dynamics and Relationship Satisfaction among LGB Adolescents: The Role of Sexual Minority Oppression"

_children, 2021, doi:10.3390/children8030231_

Round 1

Reviewer 1 Report

The reviewed paper indeed fills a gap in the literature, concerning relationship dynamics among LGBT adolescents.

The critical remark about the reviewed paper is that there is no indication of when and where the research discussed in the paper was conducted. Only from the context it can be concluded that the research was carried out in Portugal. There is a note on page 4 (line 155): “Ethical approval for this study as granted by the Ethics Committee of the University of Beira Interior, Portugal…”. On page 13 there is, in turn, a paragraph concerning the legal context of the LGBT minority in Portugal. This paragraph, however, is not tied with the entire content of the paper (lines 333–344). Nowhere it is explicitly stated that the research was carried out in Portugal.

It is rightly stated in the paper that: “A social climate that is more affirming of young same-sex couples will likely reduce the self-stigmatization and vulnerability to oppression that has been linked to lower levels of relationship satisfaction in this study” (page 13, lines 349–351). This social climate differs depending on whether we are talking about small communities (towns or rural areas), or large cities, whose inhabitants are more open and where it is easier to feel anonymous. Therefore, it is of crucial importance to point out where (in which communities) the survey was conducted. Does the majority of the 182 participants reside in a large city or of smaller towns, or small communities? Information about this would be very important for the proper interpretation of the research results. The paper deals, i.a., with the stigmatization and discrimination of the members of LGBT minority. A sense of discrimination and stigmatization may be related to the character of the community in which a person lives. It can be hypothesized that persons living in smaller communities may experience stigmatization and discrimination to a greater extent than those living in big cities. Therefore, it is important to indicate the communities in which the respondents live. Another hypothesis is that members of LGBT minority living in smaller centers may have more problems with building relationships.

In short, in my view it is needed to put more emphasis on a Portuguese context (in this regard not only legislation, but also social attitudes towards LGBT minority are important) and to point out when and where the research discussed in the paper was conducted.  

Author Response

Please find in the attachment.

Reviewer 2 Report

The abstract needs significant improvement. The first sentence regarding stigmatized minority group doesnt convey that the nature of that stigma is related to the element of identify being explored. Asian Americans are a stigmatized minority group, but they are not stigmatized, like LGB people are, concerning the subject of study that is intimate relationship satisfaction. Similarly "developing positive relationships" doesn't convey that this study concerns intimate relationship satisfaction. LGB people face a challenge to develop positive relationships with heterosexual men. That is worth study. That is not the topic here. Basically, my critique is that the first sentence alone needs greater precision of language. This is somewhat a concern throughout. The title conveys the topic much better but then I get confused as soon as I start the abstract. In second sentence "in a relationship" is not precise enough. APA style doesnt generally allow beginning sentences with a number, but maybe in abstract or this journal it is ok? 182... vs. A sample of 182. Concerning "participants showed median levels of oppression" is an odd sentence clause. I do not know what that means. median levels? does that mean all were near the median of the norm group or near the median meaning low variability within the group? oppression? Does that mean experience of oppression, specifically sexual minority oppression? or feeling oppressed? or they oppress their partner?. I think the second clause of the sentence is more key and clearer. young lbg couples maybe should be adolescent LGB couples?

The intro is pretty strong in terms of writing and setting up in more general terms a need for a study like this, but i think it must be improved as i don't see clear evidence that the three objectives are all introduced to the same level.

Materials and methods appears to have 4 paragraphs on materials and 1 on methods, which is concerned with recruitment, instructions, rewards, and ethics approval. It doesnt appear to cover all elements of methods, such as procedures (somewhat covered) but more so participants and data analysis. I see this as more an element of description rather than that the methods are likely inadequate.

The three purposes could have been better described right before methods section. they could be explicit research questions or have a better description of what i'm going to read in the results section. for example, regarding satisfaction, there were multiple comparisons with t tests that are not adequately described/implied from teh purpose statement of "assess the levels of satisfaction with their relationships". 

Were there bonferoni corrections run? the absence of a data analysis section in methods is also related to the concern about clarity of purpose/research question and the like. 

The table should be improved in presentation. hmm. Ok, the issue is that only p not t/f/df/p should be given in that part and definitely do not separte the . from the 845 to convey the p value such as seen in the first one and similar thoughout. These are the most important changes. Without seeing it, i can't tell if that is enough. maybe. but probably not.

next, the label for the whole table 4 is probably insufficient and/or layout is not clear enough. So be more precise with title and consider what info to provide in the table for the purpose at hand. be clearer. 

is that either males or females did not signficantly differe on couple's issues subscale of the oppression instrument? I guess i wonder whehter you need all those M/SD too then. i would prefer if you think you need them, then put them as a descriptives table, and then do the tests of signficance in a different table. 

in apa style there are spaces in equations and stat symbols are italicized. if the journal follows apa style please fix. 

i'm fairly comfortable with the discussion section. it is harder to critique until the abstract, introduction, research questions, methods and results sections would be cleaned up, but i'm generally comfortable there. my suggestion would be to revisit it once you clean up the earlier parts to make sure the entire manuscript is tight. Do you adequately address in discussion the 3 re-worded research questions once you have better introduced them and presented them (at least as concerns table 4)?

Author Response

Please see attached file for point-by-point replies to Reviewer #1.

English language and style were edited by a native English speaker. 

Thank You!

Reviewer 3 Report

The introduction and/or methods section should include that the sample was drawn from participants in Portugal, not just in the in discussion section. In fact, the Portuguese socio-political context of homophobia should be part of the introduction as well as the discussion. What is the legal age of consent to marry in Portugal? Is same-sex marriage legal there? You have a variable about marital status, but nowhere does it indicate if same-sex marriages are legally recognized and at what age people can marry. Considering the sample size is 16-20, these could impact findings. 

The methods section should include inclusion criteria and response rate. Were all responses included or were some excluded from the final analysis? Convenience sampling is less generalizable than probability sampling, but does have its uses in social science research. The concerns here are self-selection bias and regression fallacy. With 182 respondents, a power analysis is recommended. 

Author Response

Please see attached file for a point-by-point reply to Reviewer #2.

English language and style were revised by a Native English Speaker.

Thank you!
